# The Biological Activity of *Monarda didyma* L. Essential Oil and Its Effect as a Diet Supplement in Mice and Broiler Chicken

**DOI:** 10.3390/molecules26113368

**Published:** 2021-06-02

**Authors:** Héloïse Côté, André Pichette, Alexis St-Gelais, Jean Legault

**Affiliations:** 1Laboratoire d’Analyse et de Séparation des Essences Végétales, Université du Québec à Chicoutimi, 555 boulevard de l’Université, Chicoutimi, QC G7H 2B1, Canada; heloise.cote@uqac.ca (H.C.); Andre_Pichette@uqac.ca (A.P.); 2Centre de Recherche sur la Boréalie (CREB), Département des Sciences Fondamentales, Université du Québec à Chicoutimi, 555 boulevard de l’Université, Chicoutimi, QC G7H 2B1, Canada; 3Laboratoire PhytoChemia Inc., 628 Boul. du Saguenay Ouest, Saguenay, QC G7J 1H4, Canada; a.st-gelais@phytochemia.com

**Keywords:** *Monarda didyma*, antibacterial activity, essential oil, broiler, mouse

## Abstract

The use of growth-promoting antibiotics in livestock faces increasing scrutiny and opposition due to concerns about the increased occurrence of antibiotic-resistant bacteria. Alternative solutions are being sought, and plants of Lamiaceae may provide an alternative to synthetic antibiotics in animal nutrition. In this study, we extracted essential oil from *Monarda didyma*, a member of the Lamiaceae family. We examined the chemical composition of the essential oil and then evaluated the antibacterial, antioxidant, and anti-inflammatory activities of *M. didyma* essential oil and its main compounds *in vitro*. We then evaluated the effectiveness of *M. didyma* essential oil in regard to growth performance, feed efficiency, and mortality in both mice and broilers. Carvacrol (49.03%) was the dominant compound in the essential oil extracts. *M. didyma* essential oil demonstrated antibacterial properties against *Escherichia coli* (MIC = 87 µg·mL^−1^), *Staphylococcus aureus* (MIC = 47 µg·mL^−1^), and *Clostridium perfringens* (MIC = 35 µg·mL^−1^). Supplementing the diet of mice with essential oil at a concentration of 0.1% significantly increased body weight (+5.4%) and feed efficiency (+18.85%). In broilers, *M. didyma* essential oil significantly improved body weight gain (2.64%). Our results suggest that adding *M. didyma* essential oil to the diet of broilers offers a potential substitute for antibiotic growth promoters.

## 1. Introduction

Antibiotic growth promoters (AGPs) have long been used in animal nutrition to reduce mortality and increase the feed conversion ratio and body weight gain, especially for broilers and swine [1,2,3,4,5]. The AGPs given for animal nutrition and agents used for treatment against bacterial infection in humans often belong to the same classes of antibiotics; this heightens the risk of bacteria developing resistance to antibiotics [6]. The enhanced resistance and the issues surrounding the excessive use of in-feed AGPs have led researchers to search out more natural alternatives. In Europe, AGPs have been prohibited for use in animal nutrition since 2006 [3].

In Canada, poultry production involves more than 2600 regulated chicken producers who have access to several AGPs approved as feed additives for poultry [6]. The use of AGPs in Canada is still permitted; however, since 2014, producers have been forced to remove those antibiotics considered as the most important for human health. This important restriction is part of a strategy involving the responsible use of antibiotics by producers and the search for new alternatives. A similar strategy for controlling antibiotic resistance is used in the United States, although there are some differences in regard to their classification of antibiotics compared to the Canadian and World Health Organization (WHO) classifications.

The drive for alternatives to standard AGPs involves finding compounds that ensure poultry health and decrease bacterial resistance while maintaining producer profitability. Public pressure and concerns about food and environmental safety have led to an active search for alternative approaches that could eliminate or decrease the use of antibiotics, such as the use of essential oil, probiotics, prebiotics, and acidifiers in diets. As natural and multi-active agents, plant extracts and plant-derived products offer an interesting potential as substitutes for synthetic antibiotics and inorganic chemicals. Essential oil has some advantages over the others, by (1) improving growth performance and nutrient digestibility through antioxidative and anti-inflammatory activity, and (2) improving intestinal and general health of the broiler via antibacterial and coccidiostatic effects [7,8,9,10]. Both the food industry and animal producers have increased their interest in the use of essential oils (EOs, volatile plant compounds). The positive effects of EO in the digestive tract of animals include the stabilizing of gut microbiota; this benefit could lead to further practices that improve intestinal health, digestion, and growth performance of livestock [6].

Some EOs have already been investigated in animal nutrition studies [11,12,13,14,15,16]. Plants from the Lamiaceae family are particularly promising [15,16,17,18,19]; for example, Denli et al. [20] added thyme essential oil to the diet of quail, resulting in significantly increased body weight gain and better feed efficiency relative to a control group. A member of the Lamiaceae, *Monarda didyma* L., has been used for millennia by indigenous people in North America to treat colic, flatulence, colds, and flu. This plant is also used in infusions for intestinal and stomach problems [21], and is particularly useful for treating digestive disorders [22].

At present, there are no studies on the bioactivity of *M. didyma* EO. Nonetheless, compounds such as thymol and carvacrol—often present in the EO of Monarda species [22]—have marked antibacterial and anti-inflammatory activities [23,24,25]. Moreover, Ertas et al. [26] showed that thymol and carvacrol help stimulate digestion by affecting pathogenic microorganisms in the gut of broilers, thereby increasing their food conversion rate and weight. The protection and reinforcement of the intestinal microbiome in broilers is critical for maintaining the overall health and optimal absorption of nutrients in these animals [27].

In this study, steam distillation was used to extract *M. didyma* EO, and GC-MS and GC-FID were used to analyze the composition of the EO. The antibacterial, antioxidant, and anti-inflammatory activities of *M. didyma* EO in vitro was assessed and results compared with the positive control *Thymus vulgaris* [24]. The objective of this study was to substitute standard antibiotics with *M. didyma* essential oil, although the main compounds responsible for the biological activities were identified and the effects of *M. didyma* EO on the growth performance, feed intake, and feed efficiency of mice and broilers were then evaluated.

## 2. Results and Discussion

*M. didyma* EO, isolated by steam distillation, comprised mainly monoterpenes, which represented approximately 98% of the total, and some sesquiterpenes (1.24%). *M. didyma* EO contained a high level of carvacrol (49.03%). The other main compounds were γ-terpinene (16.90%), *p*-cymene (7.60%), thymol (6.17%), carvacrol methyl ether (4.18%), 1-octen-3-ol (3.07%), α-terpinene (2.79%), myrcene (2.33%), α-thujene (1.39%), and limonene (1.03%) (Table 1). In total, 23 compounds were identified; all compounds had already been described in the scientific literature [28].

These compound concentrations showed some differences to previous research on EO extracts from *M. didyma*. In contrast to our observation of the highest compound levels being for carvacrol [29], analysis of EO from *M. didyma* cultivated in central Italy found thymol (59.3%) to be the most dominant compound, followed by *p*-cymene (10.3%), terpinolene (9.2%), delta-3-carene (4.4%), myrcene (3.7%), and camphene (3.4%). In 1967, Scora [30], using *M. didyma* EO from California, found γ-terpinene (27.36%) as the most important compound, followed by d-limonene (12.93%), 1,8 cineole (12.24%), bornyl acetate (7.74%), ketone (5.50%), β-myrcene (4.88%), linalyl acetate (5.06%), α-terpinol (4.20%), and α-pinene (3.10%). Scora [30] did not observe thymol or carvacrol in their EO samples; they concluded that the oil from the inflorescences contained smaller amounts of terpene components. Finally, Carron et al. [31] analyzed several North American *Monarda* species. They found thymol and carvacrol to vary markedly between taxa, ranging from 8.5% thymol and 2.9% carvacrol in *M. didyma* (red bergamot variety) to 46.6% thymol and 19.2% carvacrol in *M. fistulosa* (Richters variety).

The high relative concentration of carvacrol and the presence of thymol suggested a promising antibacterial activity for *M. didyma* EO. Our observations of the antibacterial activity of *M. didyma* EO against *E. coli* (MIC = 87 µg·mL^−1^) and against *S. aureus* (MIC = 47 µg·mL^−1^) confirmed this antibacterial potential. The positive control *T. vulgaris* (Appendix A) was also effective against *E. coli* (MIC = 109 µg·mL^−1^) and *S. aureus* (MIC = 111 µg·mL^−1^). The antibacterial activity of *M. didyma* EO versus *E. coli* and *S. aureus* was provided mainly by carvacrol, which had respective MIC values of 65 µg·mL^−1^ and 36 µg·mL^−1^, and thymol, which had respective MIC values of 110 µg·mL^−1^ and 130 µg·mL^−1^. *M. didyma* EO was also active versus *C. perfringens*, producing an MIC of 35 µg·mL^−1^; this antibacterial activity versus *C. perfringens* was provided mainly by carvacrol that had an MIC of 18 µg·mL^−1^, as well as thymol, having an MIC of 55 µg·mL^−1^, and also limonene, which produced an MIC of 29 µg·mL^−1^ (Table 2).

*M. didyma* EO also demonstrated anti-inflammatory activity. A concentration of 35 µg·mL^−1^ inhibited fifty percent (IC_50_) of NO induced by LPS in RAW 264.7 macrophages. Carvacrol (IC_50_ = 22.6 µg·mL^−1^) and *p*-cymene (IC_50_ = 25.5 µg·mL^−1^) appeared to be responsible, in part, for this activity. Furthermore, a cell-based assay with an IC_50_ of 4.6 µg·mL^−1^ highlighted the EO antioxidant activity (Table 3).

The antibacterial and antioxidant activities of other members of the Lamiaceae family have been observed, principally for *T. vulgaris* and *Origanum vulgare* [32,33,34,35,36]; however, the biological activities of *M. didyma* are less well known. Fraternale et al. [22] used a DPPH and lipid peroxidation test to demonstrate the elevated antioxidant activity of *M. didyma* EO. Numerous studies have highlighted the antibacterial and antioxidant activities of EOs that possess high levels of phenolic compounds such as thymol, carvacrol, and eugenol [9,24,25]. Thymol and carvacrol appear to be particularly effective against Gram-negative bacteria, in part because these compounds act on the outer membrane, such as by provoking the release of lipopolycaccharides, increasing the permeability of the cytoplasmic membrane, and depolarizing the cytoplasmic membrane. Furthermore, hydroxyl groups are highly reactive and form hydrogen bonds with active sites of target enzymes, inactivating them, and consequently create a dysfunction or rupture of the cell membrane [25]. Similar to our study, Guimarães et al. [25] also observed high and rapid thymol and carvacrol activity against *E. coli*. Antioxidant and anti-inflammatory effects of the EO can produce a positive effect in the gastrointestinal tract.


Given that *M. didyma* EO demonstrated in vitro activities, the effectiveness of different concentrations of EO in the diet of mice on body weight gain (BW), feed intake (FI), and feed efficiency (FE) was assessed. A significant difference between the treatment groups and the negative control group for BW (*p* < 0.001), FI (*p* < 0.001), and FE (*p* < 0.001) was observed (Table 4). Supplementing the mouse diet with EO increased mouse BW throughout the experiment. EO concentrations of 0.1% *M. didyma* and 0.1% *T. vulgaris* markedly increased mouse BW (respectively 5.4% and 9.4%), compared with the untreated mice (average of 5.58 g). EO dietary supplements decreased overall FI (*p* < 0.001) for all EO concentrations, especially for *M. didyma* 0.2% (145.24 g) and *M. didyma* 0.1% (141.58 g), compared to untreated mice (164.83 g). Finally, *M. didyma* 0.1% and *M. didyma* 0.2% produced the lowest FE ratios at 23.98 and 25.13, respectively. Therefore, diets supplemented with *T. vulgaris* EO and *M. didyma* EO at levels of 0.2% and 0.1% improved mouse BW and FE compared to the negative control.


No studies have yet tested EO diet supplements in mice; however, Denli et al. [20] observed that *T. vulgaris* EO caused BW gain and improved FE for quail. Platel et al. [37] concluded that adding various spices in food enhanced either enzyme activity related to digestion or increased the secretion of bile. Yang et al. [38] observed that EO supplementation during the growth period increased lipase, trypsin, and chymotrypsin activities significantly. EO also increased the fecal digestibility of dry matter and the digestibility of ether extract, fiber, fat, ash, and protein [18]. The improvement of BW and FE in our study of mice was attributed to these processes.

When *M. didyma* EO was added to the diet of male broilers, during a growth period of 36 days, the treated broilers attained a BW of 2.65 kg with an average of 4.46 kg of FI by the animals. Control broilers with standard antibiotics weighed 2.58 kg after an average FI of 4.51 kg. Broilers fed with the EO-supplemented diet therefore showed a BW increase of 2.64% compared to control animals fed using standard antibiotics (Table 5). A significant difference was observed for BW between the control and *M. didyma* EO broilers in the first 10 days (268.5 g and 279 g, respectively). During the growth and finisher phases, broilers fed with the EO diet were significantly heavier than the control broilers with antibiotics and followed the same growth curve (Appendix A).

Existing studies identify four different mechanisms that are important for EO action—sensorial, metabolic, antioxidant, and antibacterial activities [13]. Hashemipour et al. [39] demonstrated that supplementing a broiler diet with 100 mg/kg and 200 mg/kg of thymol and carvacrol, respectively, produced no effect on FI, but significantly increased BW gain and improved the feed conversion ratio. Antibiotics in animals improve the BW by 5–6% and the FE by 3–4%, with the most pronounced effects observed in young animals [6]. Khattak et al. [15], supplementing the diets of broilers with a natural blend of EO (including thyme and oregano), observed no significant difference in growth performance during the starter period. Although higher mortality rates could be expected in the EO test group due to the lack of antibiotic use, Khattak et al. [15] found no significant difference between the mortality of broilers having the EO-supplemented diet and that of the control broilers.

## 3. Materials and Methods

### 3.1. Chemicals

Standard compounds used for GC analyses and biological testing were obtained from Sigma-Aldrich (St. Louis, MO, USA); the compounds included thymol, carvacrol, γ-terpinene, *p*-cymene, α-terpinene, myrcene, and limonene.

### 3.2. Plant Material and Extraction of Essential Oils

*M. didyma* L. EO was extracted by steam distillation for three hours from freshly harvested aerial parts (refractive index = 1.4977; density = 0.939 g·mL^−1^; yield (%) = 0.49). The plant came from Saint-Fulgence, Québec and was harvested at the beginning of August (Voucher QFA0625784). *Thymus vulgaris* L. EO (refractive index = 1.4797; density = 0.923 g·mL^−1^) was purchased from Aliksir (Grondines, QC, Canada). The EO yield from extraction depended on the total amount of the raw material. The EO was then stored in the dark at a temperature of 4 °C until needed.

### 3.3. GC-FID and GC-MS Analysis

All chromatographic analyses were run on an Agilent 6890N GC (Agilent Technologies, Santa Clara, CA, USA) equipped with a non-polar DB-5 column and a polar SolGel-Wax column (30 m × 0.25 mm × 0.25 mm) as well as two flame ionization detectors (FIDs) (Agilent Technologies, Santa Clara, CA, USA). The oils were injected in an undiluted (0.1 µL injection volume, split 1:235) and undried state (Appendix A). The temperature program began at 40 °C for 2 min, then rose 2 °C·min^−1^ up to 210 °C. The temperature was then held at 210 °C for 13 min. Samples were also injected into an Agilent 7890A GC (Agilent Technologies, Santa Clara, CA, USA) coupled to an Agilent 5975C InertXL EI/CI mass spectrometer (MS) equipped with a DB-5MS column using the same temperature program as above and a split of 1:1000. Compounds were identified from their retention indexes as calculated from even-numbered C7 to C36 alkane standards and from Wiley 6N, MS databases (NIST08) and standard injection when available (LASEVE, UQAC, Chicoutimi, QC, Canada). Quantification was derived from the FID response on the DB-5 column without any correction factor.

### 3.4. In Vitro Activity of Essential Oil

#### 3.4.1. Cell Culture

Healthy human skin fibroblasts WS1 (ATCC CRL-1502) and the murine macrophage RAW 264.7 (ATCC TIB-71) were obtained from the American Type Culture Collection (Manassas, VA, USA). Cells were grown in a humidified atmosphere at 37 °C in 5% CO_2_, in Dulbecco’s minimum essential medium supplemented with 10% fetal calf serum (Hyclone, Logan, UT, USA), 1 × solution of sodium pyruvate, 1 × vitamins, 1 × non-essential amino acids, 100 IU of penicillin, and 100 µg·mL^−1^ streptomycin (Cellgro^®^, Mediatech, Manassas, VA, USA).

#### 3.4.2. Bacterial Strains

The in vitro antimicrobial activity of *M. didyma* EO was tested against Gram-negative *Escherichia coli* (ATCC 25922), Gram-positive *Staphylococcus aureus* (ATCC 25923), and Gram-positive *Clostridium perfringens* provided by the Chicoutimi Hospital, Saguenay, Canada. Bacteria were grown in a humidified atmosphere at 37 °C.

#### 3.4.3. Culture Methods

Bacteria were stored at −80 °C until use. For culturing the bacteria, all bacteria were placed in a nutrient broth base (Difco) for 16–18 h at 37 °C; *C. perfringens* was grown in an anaerobic vial. The cellular density of the inoculum was measured via optical density at 600 nm for *E. coli* [40], 660 nm for *S. aureus* [41], and 450 nm for *C. perfringens* using a Multiskan™ GO Spectrophotometer (Thermo Fisher Scientific). The inoculum was re-diluted in the nutrient broth to obtain the required bacterial concentration.

#### 3.4.4. Measurement of Anti-Inflammatory Activity

The inhibition of nitric oxide (NO) production by *M. didyma* EO and compounds was evaluated following Legault et al. [42]. Control L-NAME was used as a positive control. The murine macrophage RAW 264.7 cells were incubated with EO or dissolved compounds in DMSO, then stimulated using 100 ng·mL^−1^ LPS and incubated at 37 °C. After 24 h, the cell-free supernatant was collected and the NO concentration immediately determined using the Griess reaction. The absorbance was read at 540 nm using an automated Varioskan Ascent plate reader, and the presence of nitrite was quantified by comparing with a NaNO^2^ standard curve, and IC_50_ expressed 50% of NO inhibition.

#### 3.4.5. Evaluation of Antioxidant Activity Using Cell-Based Assays

The antioxidant activity was evaluated using the DCFH-DA assay as described by Girard-Lalancette et al. [43]. Human skin fibroblasts WS1 were incubated for 1 h with a growing concentration of EO or dissolved compounds in DMSO. One hundred microliters of 200 µM tert-butylhydroperoxide was then added and fluorescence immediately measured, and again after 90 min. Measurements were made on an automated plate reader (Fluoroskan Ascent FL, Labsystems, Milford, MA, USA) using an excitation wavelength of 485 nm and an emission wavelength of 530 nm. Antioxidant activity was expressed as the concentration of extract inhibiting 50% (IC_50_) of DCFH oxidation.

#### 3.4.6. Evaluation of Antioxidant Activity Using ORAC

The method described by Ou et al. [44] with some modifications was followed. Briefly, the ORAC assay was carried out in black 384-well microplates (Nunc) on a Fluoroskan Ascent FL™ plate reader (Labsystems) equipped with an automated injector. Four concentrations of Trolox (the control standard) (1.56, 3.13, 6.25, and 12.5 μM) were used in quadruplicate, and a gradient of 16 concentrations of the samples (compounds dissolved in DMSO or pure EO) was prepared without replication. The experiment was conducted at 37.5 °C and in a pH 7.4 phosphate buffer with a blank sample run in parallel. The fluorimeter was programmed to record the fluorescence (λ ex.: 485 nm/em: 530 nm) of fluorescein every minute after the addition of 375 mM of 2,2′-azobis(2-amidinopropane) dihydrochloride (AAPH), for a total of 60 min. The final values were calculated using the net area under the curves of the sample concentrations for which a decrease of at least 95% of fluorescence was observed at 60 min. ORAC values were expressed in micromoles of Trolox equivalents (TE) per milligram (μmol TE·mL^−1^).

#### 3.4.7. Evaluation of Antibacterial Activity

The antibacterial activity of *M. didyma* EO and compounds was tested using the antibacterial hydrophobic assay as described by Côté et al. [45]. Briefly, after micro-organisms passed 16–18 h at 37 °C in a nutrient broth base (Difco), 20 µL methanol containing growing concentrations of EO and compounds (3.1 to 200 µg·mL^−1^) was transferred onto nutrient agar in 96-well plates. Bacterial strains having a concentration of 2.5 × 10^5^ colony-forming units (CFU) were then added per mL of nutrient broth. Bacterial suspension without treatment was used as a negative control, and bacterial suspension plus solvent were tested in parallel to demonstrate the absence of solvent toxicity. The blank consisted of a culture medium only and was subtracted from all subsequent measurements of each well. The 96-well plates were then incubated at 37 °C for 5 h to foster bacterial growth. One hundred microliters of resazurin sodium salt solution with a concentration of 50 µg·mL^−1^ (Sigma R-2127, St-Louis, MO, USA) was then added to each well. Fluorescence was read on an automated Fluoroskan Ascent FLTM plate reader (Labsystems, Milford, MA, USA) after 2 h for *S. aureus* and 3 h for *E. coli* and *C. perfringens*. The MIC was determined as the lowest concentration resulting in 95% inhibition of bacterial growth.

### 3.5. Experimental Design in Mice

Mice were fed with a mouse diet specifically designed to support the growth and maintenance of animals (LabDiet 5015; 20% protein, 25% fat, 55% carbohydrate). The treatments were prepared by adding concentrations of *M. didyma* and *T. vulgaris* (used as the positive control) EO at 0.2% and 0.1%, to the original control diet. The food containing EO was prepared every week to ensure its freshness and quality. Our experiment was assessed for its ethical acceptability and was approved by the APC (Animal Protection Committee of UQAC, Université du Québec à Chicoutimi). Mice were housed in cages in an environmentally controlled room, having 12 h of darkness followed by a period of 12 h of light during the entire run of the experiment. Each cage was provided with a single feed and water system to provide *ad libitum* access. Fifty (50) mice (five-week-old Charles River males) were placed in cages; each cage contained five randomly selected mice (two replicates). Each cage was assigned to one of five dietary treatments (four EO treatments and one negative control), giving a total of ten (10) experimental cages. The study was completed after 80 days of treatment. BW was measured twice a week and average FI determined on a weekly basis. For each week, the FE per cage was calculated based on the average BW and FI per mouse.

### 3.6. Experimental Design for Broilers

Male broilers (*n* = 8216) were divided into four treatment groups: two control (*n* = 2054 each) and two treatment (*n* = 2054 each). The broilers were installed in an insulated room with facilities for control temperature, light, and humidity according to industry standards. Broilers were fed with the Nutrinor standard diet for poultry. The feeding program consisted of a starter (1–10 d), growth (11–20 d), and finisher (21–35 d) diet given to each broiler *ad libitum*. The treatment group received 0.5% *M. didyma* EO in the starter diet, 0.1% EO in the growth diet, and 0.05% EO in the finisher diet. The control group received standard antibiotics: Maxiban^TM^ (0.5 kg/metric tons (MT)) and Tylosine (0.25 kg/MT). Broilers were weighed daily using an automatic balance installed in the hen house. Mortality was recorded daily and total FI was determined at the end of the experiment.

### 3.7. Statistical Analysis

Analysis of variance (ANOVA) was run on all our collected data using the statistical analysis system Sigma STAT. If appropriate, multiple comparison procedures were performed with the Holm–Sidak method. Statements of statistical significance are based on *p* < 0.05.

## 4. Conclusions

Our results confirmed the in vitro antibacterial, antioxidant, and anti-inflammatory activities of *M. didyma* EO. A diet supplemented with *M. didyma* EO significantly increased mouse BW and suggested good biological activity in broilers. EO has good potential as an alternative to synthetic antibiotics used in animal nutrition. Future research must explore *M. didyma* EO in vivo antibacterial activity against *E. coli* and *C. perfringens.* Nonetheless, the different dosages of *M. didyma* EO in the starter, growth, and finisher phases of the broiler diets remain to be tested to determine the optimal concentrations.

## Figures and Tables

**Table 1 molecules-26-03368-t001:** Composition of *M. didyma* L. essential oil.

Identified Compounds			Relative Concentration (%)
RI	Name	Identification ^1^	
930	α-thujene	MS, RI	1.39
938	α-pinene	MS, RI	0.43
976	1-octen-3-ol	MS, RI	3.07
992	myrcene	MS, RI	2.33
1010	α-phellandrene	MS, RI	0.36
1016	δ-3-carene	MS, RI	0.16
1022	α-terpinene	MS, RI	2.79
1030	*p*-cymene	MS, RI	7.60
1034	limonene	MS, RI	1.03
1064	γ-terpinene	MS, RI	16.90
1072	cis-sabinene-hydrate	MS, RI	0.25
1094	linalool	MS, RI	0.15
1174	terpinen-4-ol	MS, RI	0.11
1185	α-terpineol	MS, RI	0.79
1232	thymol methyl ether	MS, RI	0.86
1248	carvacrol methyl ether	MS, RI	4.18
1294	thymol	MS, RI	6.17
1306	carvacrol	MS, RI	49.03
1435	β-caryophyllene	MS, RI	0.89
1488	germacrene D	MS, RI	0.12
1511	γ-cadinene	MS, RI	0.23
	Total		98.84

^1^ MS: Identification by GC-MS; RI: Compounds were identified by comparison of GC retention indices relative to retention times of a series of n-alkanes (C7–C36) and compared with literature data. Compounds ≤ 0.1% are not reported.

**Table 2 molecules-26-03368-t002:** Antibacterial activity of *M. didyma* and *T. vulgaris* (as positive control) essential oils and the main compounds against selected bacterial strains.

Compounds	Antibacterial Activity MIC (µg·mL^−1^)		
	*E. coli*	*S. aureus*	*C. perfringens*
*T. vulgaris* EO	109 ± 10	111 ± 7	ND
*M. didyma* EO	87 ± 8	47 ± 8	35 ± 4
Thymol	110 ± 7	130 ± 10	55 ± 3
Carvacrol	65 ± 5	36 ± 1	18 ± 1
α-terpinene	˃200	˃200	ND
γ-terpinene	˃200	˃200	ND
*p*-cymene	˃200	˃200	ND
Limonene	˃200	˃200	29 ± 1
Myrcene	˃200	˃200	ND

ND: Not determined; data are representative of three different experiments; mean ± standard deviation, *n* = 3; MIC is defined as the lowest concentration inhibiting 95% of bacterial growth.

**Table 3 molecules-26-03368-t003:** Antioxidant and anti-inflammatory activity of *M. didyma* and *T. vulgaris* (as positive control) essential oils and the main compounds.

Compounds	Antioxidant		Anti-Inflammatory
	Cell-Based Assay	ORAC	IC_50_ (µg·mL^−1^)
	IC_50_ (µg·mL^−1^)	µmol Trolox·mg^−1^	
*T. vulgaris* EO	11 ± 9	0.4 ± 0.2	64 ± 6
*M. didyma* EO	4.6 ± 0.3	0.52 ± 0.01	35 ± 4
Thymol	˃200	1.34 ± 0.03	˃200
Carvacrol	54 ± 9	2.3 ± 0.3	22.6 ± 0.2
α-terpinene	3.4 ± 0.2	0.17 ± 0.05	˃200
γ-terpinene	ND	ND	˃200
*p*-cymene	˃200	0.02 ± 0.01	25.5 ± 0.9
Limonene	6.2 ± 0.5	0.08 ± 0.02	22 ± 7
Myrcene	92 ± 4	0.04 ± 0.01	˃200
Trolox	ND	9 ± 1	ND

ND: Not determined; data are representative of three different experiments; mean ± standard deviation, *n* = 3.

**Table 4 molecules-26-03368-t004:** Comparison of mean body weight gain (g), feed intake (g), and feed efficiency of the mouse treatments.

Treatments	Body Weight Gain (g) (BW)	Feed Intake (g) (FI)	Feed Efficiency (FI/BW)
	0–80 Days	0–80 Days	0–80 Days
Control	5.58 ± 1.01 ^a^	164.83 ± 1.55 ^a^	29.55 ^a^
*T. vulgaris* 0.2%	5.56 ± 1.41 ^b^	154.96 ± 0.87 ^b^	27.87 ^b^
*T. vulgaris* 0.1%	6.16 ± 1.44 ^b^	163.39 ± 14.21 ^c^	26.54 ^c^
*M. didyma* 0.2%	5.78 ± 1.40 ^c^	145.24 ± 7.81 ^c^	25.13 ^c^
*M. didyma* 0.1%	5.90 ± 1.17 ^c^	141.58 ± 2.78 ^c^	23.98 ^c^

Values with different letters in the same column (a–c) differ significantly (two-way analysis of variance, *p* < 0.05).

**Table 5 molecules-26-03368-t005:** Comparison of mean body weight gain (g), feed intake (g), and feed efficiency of the broiler treatments.

Treatments	Body Weight Gain (g) (BW)		Feed Intake (g) (FI)		Feed Efficiency (FI/BW)	
	0–10 Days	0–36 Days	0–10 Days	0–36 Days	0–10 Days	0–36 Days
Antibiotics suppl.	269 ± 7 ^a^	2578 ± 117 ^a^	384.71 ^¥^	4514.25 ^¥^	1.43 ^¥^	1.75 ^¥^
*M. didyma* suppl.	279 ± 8 ^b^	2652 ± 121 ^b^	360.36 ^¥^	4455.81 ^¥^	1.29 ^¥^	1.68 ^¥^

^¥^ Values are available only at the end of the experiment; values in the same column with different letters (a–b) differ significantly (two-way analysis of variance, *p* < 0.05).

## Data Availability

The data presented in this study are available in Appendix A.

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
