# Peer review of "The Biological Activity of Monarda didyma L. Essential Oil and Its Effect as a Diet Supplement in Mice and Broiler Chicken"

_molecules, 2021, doi:10.3390/molecules26113368_

Round 1

Reviewer 1 Report

Dear authors, your work is clear and well done. I think that every part is complete and well supported.

I would like to give some little suggestions:

  • at the end of the introduction please add a little sentence in which you declare the objectives of your work. This can make more understandable the overall work.
  • in the results and discussion section, please specify the meaning of BW, FI and FE. They are at page 5 and this is the first appearance. Furthermore I suggest to add some graphics for the body weigh growth during the treatments (put them in supplementary section if there is no space). 
  • in materials and methods section, part 3.2, I think it's better if you say how much is the yield of EO extraction. Then in 3.3, the two columns work in series? In 3.4 I suggest to add that the human cell line WS1 is from fibroblasts, add the code or the origin of C. perfringens and the temperature of C. perfringens growth (same as E. coli and S. aureus?)
  • last suggestion is to delete eventual - that are present in the text

Reviewer 2 Report

The author studied biological activity of Monarda didyma and its effect as a diet supplement. The experiments performed logically and its results were described well.

However, there are some minor questions for publication.

  1. Chemical analysis of EO
  2. Statistical analysis

Thymus vulagris as positive control.

The author used T. vulagris EO as positive control for antibacterial, antioxidant, anti- inflammatory activity. However, Denli et al. did not test such activity. Therefore, the EO could not be used as positive control. Find the references that T. vulagris EO had antibacterial, antioxidant, anti-inflammatory activities and cited them, or find a suitable EO as positive control.

Table 1. add the composition of T. vulagris EO.

Because composition of EO changed according to its origin, harvest time and part and so on. Therefore, it would better to describe composition of T. vularis EO for comparison with M. didyma EO.

RI: If RI values are obtained from the experiment, then describe how to obtain. Eg.) Compounds were identified by comparison of GC retention indices relative to retention times of a series of n-alkanes(C7-C25) with those reported in the literature (Adams.)

Or IfRI values are obtained from literature, then indicate RI was obtained from literature and change RI in identification to literature.

Describe what “–“ represents. If camphene and D-cardiene were not detected, delete them.

D-cardiene: change D to small capital D,

  • erpineole: change terpineole to terpineol

ρ-cymene: change ρ-cymene to p-cymene, it is para-cymene.

<Statistical analysis>

Table 4 and 5.: The authors described that the results were obtained from tw0-way analysis of variance, two-way ANOVA. What were the factors, and what about the interaction of two factors? Please indicated the factors to be analyzed and describe the statistical values in context.

Table 4. The authors also indicated that values with different letteres(a-f) differ significantly.

  1. There were no d,e,f in table 4
  2. Body weight gain 5.58 and feed intake 164.83, and feed efficiency 29.55 were all “a” that meant all the values are not statistically If so, it all values are looked like no significantly difference. Please re-check the statistical analysis. Whether it was analyzed by one-way ANOVA, and values with different letters “in columns” differ significantly.

Indicate how to calculated IC50 values in statistical analysis, M&M section and describe 90% Confident limit.

Materials and methods

Indicate the place and time which the EO was obtained (Monarda didyma) Indicate the origin (place) of Thymus vulgaris EO

<Chemical analysis>

GC-MS: The samples for GC-MS was 5 uL of EO in 500 uL of Hexane, that is about 10,000 ppm, and injected 3 uL as splitless mode. That means 30 ug of EO was injected into GC column.

Usually, for EO analysis, we prepared 5,000 ppm and injected 1 uL as split mode (1:50 -1:20), that, 100-250 ng of EO was injected into GC column. It gave good resolution.

According to your GC-MS method, 120-300-fold much amount was injected than our method. It would be overlap of chemical components, and led to bad resolution of EO.

Please re-check the GC-MS analysis method. If it is corrected, please add the figure of analysis.

GC oven: 40C for 2 min, gradually raised to 210C at 2C/min for 33 min. What does 33 min mean? Is it meant temperature was hold 33 min at 210C?

If so, describe as 40C for 2 min, gradually raised to 210C at 2C/min, and hold that temperature for 33 min.

If it indicated total analysis time is 33 min, it should be re-check. The total time is 87 min.

<Minor Points: Typo error>

Page 2. Last paragraph, d-3-carene: change d to delta.

Delta-limonene: change delta to D(small capital) Page 5. Change zSimilar to Similar.

Italicize all scientific name through the manuscript

Reviewer 3 Report

The manuscript describes the evaluation of essential oil (EO) extracted from Monarda didyma L. as an alternative growth promoter to antibiotics. The composition of M. didyma EO was identified and the EO’s antibacterial, antioxidant and anti-inflammatory activities were assessed. Finally, the growth-promoting efficacies of M. didyma EO were evaluated in murine and broiler models. Overall, the manuscript has soundly described and demonstrated M. didyma EO as a potential growth promoter and is considered a novel alternative for antibiotic growth promoters. The reviewer recommends the manuscript be published after minor revision. The specific comments are listed below.

  1. In the introduction section, the authors are recommended to also describe the other alternatives growth promoters that are being explored in the field e.g. probiotics, prebiotics, and acidifiers. Then explain the motivation for the current investigation of M. didyma EO, compared to the other growth promoter alternatives. Is there a potential for cost-effectiveness or other reasons?

  1. Regarding the experimental method for broilers, in parallel to the described M. didyma EO treatment, the authors are recommended to describe the antibiotic treatment group such as the name of antibiotic used, and % of antibiotic added in starter, growth, and finisher phase.

  1. For a better representation of the data, the authors are recommended to express the body weight gain, feed intake, and feed efficiency as mean ± standard deviation in table 4 and table 5.

  1. In table 4, the authors are advised to clarify the following statement. “Values with different letters (a–f) differ significantly (two-way analysis of variance, p < 0.05).” Are there any differences in the p values between the a-b groups and a-c groups?

  1. The authors proposed future M. didyma EO evaluation against Eimeria sp. in the conclusion. The authors are advised to give the reason for including Eimeria sp. to give a better context for the readers.

  1. The last two sentences of the conclusion section: “Nonetheless, the mode of action of M. didyma EO remains to be specified, and optimal dosages of M. didyma EO in the starter, growth, and finisher phases of broiler diet remain to be fixed” were hard to understand, particularly the underlined phrases. The authors are recommended to revise the sentences.

  1. The superscripts and subscripts used throughout the main text, especially in the materials and method section should be revised, e.g. NaNO2, mL-1, and x 103 CFU.

Reviewer 4 Report

The manuscript molecules-1157792 entitled “The biological activity of Monarda didyma L. essential oil and its effect as a diet supplement in mice and broiler chicken” reports the investigation on the biological activities of the EO of Monarda didyma. Particularly important are the results on the BW gain in mice and chicken broilers in virtue of the demanding for reducing the use of antibiotics in animal production.

The manuscript shows several drawbacks that need to be amended.

Abstract: the parameters indicated: Escherichia coli (IC90 = 66 μg·mL−1_), Staphylococcus aureus (IC90 = 33 μg·mL−1_), and Clostridium perfringens (IC90 = 20 μg·mL−1_) does not exist in Microbiology. The parameters are Minimum Inhibitory Concentration and the Minimum Bactericidal Concentration. The MIC90 that the authors also indicate is another parameter that is incorrectly used here.

MIC90 is used in Clinical Microbiology to indicate that 90% of the known clinical strains show that MIC value. The same for MIC50: indicates that 50 % of the known clinical strains show that MIC value.

The MIC value is the minimum concentration that inhibits the 95-100% of the bacterial growth, and the MBC value is the minimum concentration that eliminates the bacterial growth (no recovery of bacterial cells at that minimum concentration).

Change this all over the manuscript, particular attention to page 3 and page 4

Page 4 the indicated MIC50 after Table 3 does not make sense!

Page 2 change microflora to microbiota

Page 2 where is …no studies of the bioactivity change to no studies on the bioactivity

Page 2 where is  “In this study, we use steam…..” change to In this study, we used

 steam distillation to extract M. didyma EO, and we runned GC-MS and GC-FID to analyze the composition of the EO.

Page 2 and all over the manuscript the reviewer stated that the authors focus on themselves instead on the research by the incredible overused of the first person “we”. The focus is the research and not the persons who conducted the research. Change all over the manuscript: e.g. Page 2 where is:” We assess the antibacterial, antioxi-dant, and anti-inflammatory activities of M. didyma EO in vitro and compare our results with the positive control Thymus vulgaris [20]. We identify the main compounds respon-sible for the biological activities and then evaluate the effects of M. didyma EO on the growth performance, feed intake, and feed efficiency of mice and broilers.” Change to:

The antibacterial, antioxidant, and anti-inflammatory activities of M. didyma EO were evaluated, and the results were compared with the positive control Thymus vulgaris[20].The main compounds responsible for the biological activities were identified and the effects of M. didyma EO on the growth performance, feed intake, and feed efficiency of mice and broilers were estimated.

Page 3 The differences in the composition of the same EO according to locations must be justified. Nowadays the basis for these differences is plenty recognized and is mandatory to report them.

Page 5  change gram to Gram (this name is after Christian Gram (1853–1938))

Page 5 The authors need to review this statement: the permeability of the cytoplasmic membrane to ATP,

Page 5 the sentence “Xu et al. [25] also observed thymol and carvacrol activity against E. coli.” Is really very poor in virtue to what is known about the antibacetrial action of thymol and carvacrol!

Page 5 The authors are advised to revise this sentence ” Finally, M. didyma 0.1% and M. didyma 0.2% produced the highest FE ratios at 23.98 and 25.13, respectively.” This does not seem correct.

Check the names of the plants; they must be in italic

Section MM, 3.2 - More detail on the plant material is required, namely what part of the plant was used, the location of harvest, date of collection and where the voucher

specimens were deposited.

What strain of C. perfringens was used?

Why different l were used for the different bacteria?

What bacterial concentration was used?

Change NaNO2 to NaNO2

As the authors well pointed out there are relevant differences in the composition of the EO of the same plant grown in different regions. Why not use the main component of the EO, carvacrol instead of the main EO?

Round 2

Reviewer 4 Report

The manuscript molecules-1157792 entitled “The biological activity of Monarda didyma L. essential oil and its effect as a diet supplement in mice and broiler chicken” is now in conditions to be published.